



# Ocean acidification reduces growth and grazing of Antarctic heterotrophic nanoflagellates

Stacy Deppeler[1,2], Kai G. Schulz[3], Alyce Hancock[1,4,5], Penelope Pascoe[6], John McKinlay[6], and Andrew Davidson[5,6]

[1]National Institute of Water and Atmospheric Research, Wellington, New Zealand
[2]Institute for Marine and Antarctic Studies, University of Tasmania, Hobart, Tasmania, Australia
[3]Centre for Coastal Biogeochemistry, Southern Cross University, East Lismore, New South Wales, Australia
[4]Antarctic Gateway Partnership, Hobart, Tasmania, Australia
[5]Antarctic Climate and Ecosystems Cooperative Research Centre, Hobart, Tasmania, Australia
[6]Australian Antarctic Division, Department of the Environment and Energy, Kingston, Tasmania, Australia

**Correspondence:** Stacy Deppeler (stacy.deppeler@niwa.co.nz)

**Abstract.** High-latitude oceans have been identified as particularly vulnerable to ocean acidification if anthropogenic $CO_2$ emissions continue. Marine microbes are an essential part of the marine food web and are a critical link in biogeochemical processes in the ocean, such as the cycling of nutrients and carbon. Despite this, the response of Antarctic marine microbial communities to ocean acidification is poorly understood. We investigated the effect of increasing $f CO_2$ on the growth of

heterotrophic nanoflagellates (HNF), nano- and picophytoplankton, and prokaryotes in a natural coastal Antarctic marine microbial community from Prydz Bay, East Antarctica. At $CO_2$ levels ≥634 µatm, HNF abundance was reduced, coinciding with significantly increased abundance of picophytoplankton and prokaryotes. This increase in picophytoplankton and prokaryote abundance was likely due to a reduction in top-down control of grazing HNF. Nanophytoplankton abundance was significantly elevated in the 634 and 953 µatm treatments, suggesting that moderate increases in $CO_2$ may stimulate growth. Changes in

predator-prey interactions with ocean acidification could have a significant effect on the food web and biogeochemistry in the Southern Ocean. Based on these results, it is likely that the phytoplankton community composition in these waters will shift to communities dominated by prokaryotes, nano- and picophytoplankton. This may intensify organic matter recycling in surface waters, leading to a decline in carbon flux, as well as a reducing the quality and quantity of food available to higher trophic organisms.

## 1 Introduction

Oceanic uptake of anthropogenic $CO_2$ has resulted in a ~0.1 unit decline in pH in the oceans since pre-industrial times (Sabine, 2004; Raven et al., 2005), with ~40% of this uptake occurring in the Southern Ocean (Takahashi et al., 2012; Frölicher et al., 2015). In addition, the low overall water temperature and naturally low $CaCO_3$ saturation state make the Southern Ocean particularly vulnerable to ocean acidification (Orr et al., 2005; McNeil and Matear, 2008). Coastal Antarctic waters are regions

of high productivity, that provide an essential food source for the abundance of life in Antarctica (Arrigo et al., 2008). While large phytoplankton, such as diatoms and dinoflagellates, are often believed to be responsible for most of the energy transfer





to higher trophic levels in this region, picophytoplankton, prokaryotes, mixotrophic phytoflagellates, microheterotrophs, and heterotrophic nanoflagellates (HNF) also play important roles in grazing and the carbon cycle (Azam et al., 1991; Sherr and Sherr, 2002; Smetacek et al., 2004).

Marine microbes are an essential part of the marine food web and are a critical link in biogeochemical processes, such as the
cycling of nutrients and carbon (Azam and Malfatti, 2007). Globally, it is estimated that ∼80-100% of daily primary production is either consumed by grazers or lost via processes such as cell lysis and sinking (Behrenfeld, 2014). Grazing can profoundly affect phytoplankton abundance in marine ecosystems, with microzooplankton consuming on average 60-75% of daily primary production (Landry and Calbet, 2004) and HNF grazing between 20-100% of daily bacterial production (Pearce et al., 2010; Safi et al., 2007). Prokaryotes salvage dissolved organic matter released from phytoplankton primary production, whish is
returned to the food web upon grazing by HNF (Pearce et al., 2010; Buchan et al., 2014). Prokaryotes also produce essential micronutrients and vitamins required for phytoplankton growth (Azam and Malfatti, 2007; Buchan et al., 2014; Bertrand et al., 2015) and are important in the supply of nutrients to microzooplankton in Antarctic waters over winter, when primary productivity is low (Azam et al., 1991). This transfer of organic matter between primary producers, prokaryotes (bacteria and Archaea), and protozoa forms the microbial loop, upon which all life in the ocean relies (Azam et al., 1983; Fenchel, 2008).

In Antarctic waters, heterotrophic flagellates make a significant contribution to the top-down control of phytoplankton and prokaryote productivity. They can achieve growth rates that exceed that of their phytoplanktonic prey and their grazing can significantly alter the microbial community composition (Bjørnsen and Kuparinen, 1991; Archer et al., 1996; Pearce et al., 2010). Heterotrophic flagellates, microzooplankton, and ciliates of all sizes (2->200 μm) have been observed grazing on picophytoplankton (0.2-2 μm) and prokaryotes (0.1-5 μm) (Safi et al., 2007). Despite their importance in marine ecosystems,
they remain relatively unstudied (Caron and Hutchins, 2013). Difficulties in identification of HNF in natural seawater samples has no doubt contributed to the scarcity of published studies (Rose et al., 2004). Of the few studies that have included heterotrophic flagellates, most studies have focused on the larger microzooplankton community (20-200 μm), reporting no changes in abundance or grazing rates with elevated $CO_2$ (Suffrian et al., 2008; Aberle et al., 2013; Davidson et al., 2016). However, ocean acidification effects on microzooplankton grazers may also be indirect, due to changes in the abundance and
composition of their prey (Rose et al., 2009b). Thomson et al. (2016), in their Antarctic minicosm study, reported a negative effect of ocean acidification on HNF abundance when $CO_2$ concentrations were ≥750 μatm. Species-specific responses to ocean acidification have also been observed amongst choanoflagellates in the present study (Hancock et al., 2018), exposing a hitherto unrecognised layer of complexity to predicting the effects of ocean acidification on microbial communities.

When assessing ocean acidification studies globally, Schulz et al. (2017) reported a general trend toward increased abundance
of picophytoplankton with declining ocean pH. The cyanobacterium *Synechococcus* and picoeukaryotes in the prasinophyte class were identified as the key beneficiaries of increased $CO_2$ levels, potentially through increased $CO_2$ concentration in the relatively small diffusive boundary layer of these small cells, allowing for down regulation of energetically costly $CO_2$ and $HCO_3^-$ transporters into the cell (Beardall and Giordano, 2002). Unlike temperate oligotrophic ecosystems, cyanobacteria are rare in Antarctic waters (Wright et al., 2009; Lin et al., 2012; Flombaum et al., 2013; Liang et al., 2016) meaning the
picophytoplankton in waters south of the Polar Front are composed largely of eukaryotes. This group can comprise up to 33%



of total phytoplankton biomass (Wright et al., 2009; Lin et al., 2012). A minicosm study on natural communities of coastal Antarctic marine microbes observed an increase in picoeukaryote abundance at $CO_2$ levels above 750 µatm, although their results suggested that this may have been due to a reduction in top-down control of the HNF community, as opposed to a direct promotion of picoeukaryote growth (Thomson et al., 2016).

In natural marine microbial communities, prokaryotes have been shown to have a high tolerance to ocean acidification, with little effect on abundance or productivity (Grossart et al., 2006; Allgaier et al., 2008; Paulino et al., 2008; Wang et al., 2016). Prokaryote abundance and production is generally linked to increased primary production, with peaks in abundance often occurring immediately after the peak of a phytoplankton bloom (Pearce et al., 2007; Buchan et al., 2014). This is likely due to increased availability of dissolved organic matter, released by phytoplankton during growth, viral lysis, or bacterial

degradation of dead cells (Azam and Malfatti, 2007). A $CO_2$-induced increase in the production of organic matter and the formation of transparent exopolymer particles has been reported in a natural community Endres et al. (2014). This promoted bacterial abundance and stimulated enzyme production for organic matter degradation, suggesting that ocean acidification may increase the flow of carbon through the microbial loop in surface waters Endres et al. (2014). Shifts in prokaryote community composition have also been reported, although with no significant effect on total prokaryote abundance (Roy et al., 2013;

Bergen et al., 2016; Zhang et al., 2013). Instead, the composition and abundance of prokaryote communities appear to be indirectly affected by ocean acidification by altering biotic factors that influence their growth and mortality.

In our study, a natural community of marine microbes from Prydz Bay, East Antarctica was exposed to increasing levels of $CO_2$, up to 1641 µatm, in 650 L minicosms. The abundance of HNF, nano- and picophytoplankton, and prokaryotes was measured and the results used to assess whether interactions between these communities could be inferred. A previ-

ous community-level study in the Antarctic reported a decline in HNF abundance and an increase in picophytoplankton and prokaryotic abundance when $CO_2$ concentrations were $\geq$750 µatm (Davidson et al., 2016; Thomson et al., 2016; Westwood et al., 2018). We used a similar experimental design to Thomson et al. (2016) but added an initial $CO_2$ acclimation period at low light to determine whether this acclimation would alter the response previously reported.

## 2   Methods

### 2.1   Minicosm

A natural microbial assemblage from Prydz Bay, Antarctica was incubated in six 650 L polythene tanks (minicosms) and exposed to six $CO_2$ treatments; ambient (343 µatm), 506, 634, 953, 1140, and 1641 µatm. Before commencement of the experiment, all minicosms were acid washed with 10% vol:vol AR HCl, rinsed thoroughly with MilliQ water, and finally rinsed with seawater from the sampling site. Seawater to fill the minicosms was collected from amongst the decomposing

fast ice in Prydz Bay at Davis Station, Antarctica (68° 35' S 77° 58' E) on 19th November, 2014. A 7000 L polypropylene reservoir tank was filled by helicopter, using multiple collections in a thoroughly rinsed 720 L Bambi bucket. The seawater was then gravity fed from the reservoir to the minicosms through Teflon-lined hose, fitted with a 200 µm pore size Arkal filter to exclude metazooplankton that would significantly graze the microbial community. Microscopic analysis showed that very





few metazooplankton and nauplii passed through the pre-filter and they were seldom observed throughout the experiment (see Hancock et al., 2018). Thus, it is unlikely that their grazing effected the $CO_2$-induced trends in community composition in our study. All minicosms were filled simultaneously to ensure uniform distribution of microbes.

The six minicosms were housed in a temperature-controlled shipping container, with the water temperature in each mini-
cosm maintained at $0.0 \pm 0.5$ °C. The temperature in each minicosm was maintained by offsetting the cooling of the shipping container against warming of the tank water with two 300 W Fluval aquarium heaters connected via Carel temperature controllers and a temperature control program. Each minicosm was sealed with an acrylic lid and the water was gently mixed by a shielded high-density polyethylene auger, rotating at 15 rpm.

Minicosms were illuminated by two 150 W HQI-TS (Osram) metal halide lamps on a 19:5 h light:dark cycle. Low intensity
light ($0.9 \pm 0.22$ µmol photons m$^{-2}$ s$^{-1}$) was provided for the first 5 d to slow phytoplankton growth while the $CO_2$ levels were gradually raised to the target concentration for each minicosm (see below). Following this 5 d $CO_2$ acclimation period, light was progressively increased over 2 d to a final light intensity of $90.5 \pm 21.5$ µmol photons m$^{-2}$ s$^{-1}$. The microbial assemblages were then incubated for 10 d with samples taken at regular intervals (see below) and no further addition of seawater or nutrients. For further details on minicosm setup see Deppeler et al. (2018).

## 2.2   Carbonate chemistry calculation and manipulation

Carbonate chemistry was measured throughout the experiment, allowing the fugacity of $CO_2$ ($f CO_2$) to be manipulated to the desired values over the first 5 d of acclimation and then maintained for the remainder of the experiment. Samples were taken daily from each minicosm in 500 mL glass-stoppered bottles (Schott Duran) following the guidelines of Dickson et al. (2007), with sub-samples for dissolved inorganic carbon (DIC, 50 mL glass-stoppered bottles) and pH on the total scale (pH$_T$, 100
mL glass stoppered bottles) gently pressure filtered (0.2 µm) following Bockmon and Dickson (2014). For each minicosm, DIC was measured in triplicate by infrared absorption on an Apollo SciTech AS-C3 analyser equipped with a Li-cor LI-7000 detector calibrated with five prepared sodium carbonate standards (Merck Suprapur) and daily measurements of a certified reference material batch CRM127 (Dickson, 2010). DIC measurements were converted to µmol kg$^{-1}$ using calculated density from known sample temperature and salinity.

Measurements of pH$_T$ were performed using the pH indicator dye m-cresol purple (Acros Organics) following Dickson et al. (2007) and measured by a GBC UV-vis 916 spectrophotometer at 25 °C in a 10 cm thermostated cuvette. A syringe pump (Tecan Cavro XLP 6000) was used for sample delivery, dye addition, and mixing to minimise contact with air. An offset for dye impurities and instrument performance (+0.003 pH units) was determined through measurement of pH$_T$ of CRM127 and comparison with the calculated pH$_T$ from known DIC and total alkalinity (TA), including silicate and phosphate. Salinity was
measured in situ using a WTW197 conductivity meter and used with measured DIC and pH$_T$ to calculate practical alkalinity (PA) at 25 °C, using the dissociation constants for carbonic acid determined by Mehrbach et al. (1973) and Lueker et al. (2000). Total carbonate chemistry speciation was then calculated for in situ temperature conditions from measured DIC and calculated PA.



During the acclimation period, the $f\mathrm{CO_2}$ in each minicosm was adjusted daily in increments until the target level was reached, after which $f\mathrm{CO_2}$ was kept as constant as possible for the remainder of the experiment. Twice-daily measurements of pH were performed in the morning (before sampling) and the afternoon using a portable, NBS-calibrated probe (Mettler Toledo) to determine the amount of DIC to be added to the minicosm. Adjustment of the $f\mathrm{CO_2}$ in each minicosm was performed by

addition of a calculated volume of 0.2 μm filtered $\mathrm{CO_2}$-saturated natural seawater to 1000 mL infusion bags and drip-fed into the minicosms at ~50 mL min$^{-1}$. One minicosm was maintained close to the $f\mathrm{CO_2}$ of the initial (ambient) sea water (343 μatm) and was used as the control treatment, against which the effects of elevated $f\mathrm{CO_2}$ were measured. The mean $f\mathrm{CO_2}$ levels in the other five minicosms were 506, 634, 953, 1140, and 1641 μatm. For further details of the carbonate chemistry sampling methods, calculations, and manipulation see Deppeler et al. (2018).

## 10    2.3    Nutrient analysis

Concentrations of the macronutrients nitrate plus nitrite (NOx), soluble reactive phosphorus (SRP), and molybdate reactive silica (silicate) were measured in each minicosm during the experiment. Samples were taken on days 1, 3, and 5 during the $\mathrm{CO_2}$ acclimation period and every 2 days for the remainder of the experiment (days 8-18). Samples were obtained following the protocol of Davidson et al. (2016). Briefly, seawater samples were filtered through 0.45 μm Sartorius filters into 50 mL

Falcon tubes and frozen at -80 °C for analysis in Australia. Determination of the concentration of NOx, SRP, and silicate were performed by Analytical Services Tasmania, using flow injection analysis.

## 2.4    Flow Cytometry

Flow cytometric analyses were performed daily to determine the abundance of small protists (HNF, pico- and nanophytoplankton, and prokaryotes) in each minicosm during the experiment. Samples were pre-filtered through a 50 μm mesh (Nitex), stored

in the dark at 4 °C, and analysed within 6 h of collection, following Thomson et al. (2016). Samples were analysed using a Becton Dickinson FACScan or FACSCalibur flow cytometer fitted with a 488 nm laser. MilliQ water was used as sheath fluid for all analysis. The analysed volume for each flow cytometer was calibrated to the sample run time and flow rate and was used to calculate final cell concentrations from event counts on bivariate scatter plots. PeakFlow Green 2.5 μm beads (Invitrogen) were added to samples as an internal fluorescence and size standard.

## 25    2.4.1    Pico- and nanophytoplankton abundance

Three pseudoreplicate 1 mL samples for pico- and nanophytoplankton abundance were prepared from each minicosm seawater sample. Each sample was placed in a beaker of ice and run for 3 min at a high flow rate of ~40 μL min$^{-1}$ for FACScan and ~70 μL min$^{-1}$ for FACSCalibur, resulting in an analysed volume of 0.1172 and 0.2093 mL, respectively. Phytoplankton populations were separated into regions based on their chlorophyll autofluorescence in bivariate scatter plots of red (FL3)

versus orange fluorescence (FL2) (Fig. 1a). The pico- and nanophytoplankton communities were determined from relative cell





size in side scatter (SSC) versus FL3 fluorescence bivariate scatter plots (Fig. 1b). Final cell counts in cells $\mathrm{L}^{-1}$ were calculated from event counts in the phytoplankton regions and analysed volume.

### 2.4.2 Heterotrophic nanoflagellate abundance

Heterotrophic nanoflagellate (HNF) abundance was determined using LysoTracker Green (Invitrogen) staining following the

protocol of Thomson et al. (2016). A 1:10 working solution of LysoTracker Green was prepared daily by diluting the commercial stock into 0.22 µm filtered seawater. For each minicosm sample, 10 mL of seawater was stained with 7.5 mL of working solution to a final stain concentration of 75 nM. Stained samples were then incubated in the dark on ice for 10 min. Triplicate 1 mL sub-samples were taken from the stained sample and run for 10 min at a high flow rate of ~40 µL $\mathrm{min}^{-1}$ for FACScan and ~70 µL $\mathrm{min}^{-1}$ for FACSCalibur, resulting in an analysed volume of 0.4153 and 0.7203 mL, respectively.

LysoTracker Green stained HNF abundances were determined in green fluorescence (FL1) versus forward scatter (FSC) plots after removal of phytoplankton and detritus particles following Rose et al. (2004) and Thomson et al. (2016) and shown in Fig. 2. Phytoplankton were identified by high chlorophyll autofluorescence in bivariate scatter plots of FL3 versus FL2 fluorescence (Fig. 2a) and detritus was identified by high SSC in FL1 fluorescence versus SSC plots (Fig. 2b). HNF abundance was then determined in a bivariate plot of FL1 fluorescence versus FSC with phytoplankton and detritus particles removed. Remaining

particles larger than the 2.5 µm PeakFlow Green beads were counted as HNF (Fig. 2c). Final cell counts in cells $\mathrm{L}^{-1}$ were calculated from event counts and analysed volume.

### 2.4.3 Prokaryote abundance

Samples for prokaryote abundance were stained for 20 min with 1:10,000 dilution SYBR Green I (Invitrogen) following Marie et al. (2005). Three pseudoreplicate 1 mL samples were prepared from each minicosm seawater sample and were run for 3

min at a low flow rate (~12 µL $\mathrm{min}^{-1}$), resulting in an analysed volume of 0.0260 and 0.0491 mL on the FACScan and FACSCalibur, respectively. Prokaryote abundance was determined from SSC versus FL1 fluorescence bivariate scatter plots (Fig. 3). Final cell counts in cells $\mathrm{L}^{-1}$ were calculated from event counts and analysed volume.

### 2.5 Statistical analysis

Microbial community growth in the minicosms was measured in six unreplicated $f\mathrm{CO}_2$ treatments and thus, sub-samples from

individual minicosms represent within-treatment pseudoreplicates. Therefore, means and standard error of these pseudoreplicate samples only provide the within-treatment sampling variability for each procedure. For the purpose of analysis, we treated pseudoreplicates as independent to provide an informal assessment of the difference among treatments. A curved (quadratic) regression model was fitted to each $\mathrm{CO}_2$ treatment over time for all analyses using the *Stats* package in R (R Core Team, 2016), with an omnibus test of differences between the trends in $\mathrm{CO}_2$ treatments over time assessed by ANOVA. Growth rates were

calculated from linear regression on the region that marked steady-state logarithmic growth and the differences between the trends in $\mathrm{CO}_2$ treatments over time was assessed by ANOVA. For peak abundance measurements, differences between treat-



ments were tested by one-way ANOVA, followed by a post-hoc Tukey test to determine which treatments differed. The lack of replication in our study and limited number of time points at which each minicosm was sampled means that the trends within treatments are indicative and the statistical differences among treatments should be interpreted conservatively. The significance level for all tests was set at <0.05.

# 3 Results

## 3.1 Carbonate chemistry

The carbonate chemistry of the initial seawater was measured as a $pH_T$ and DIC of 8.08 and 2187 µmol kg$^{-1}$, respectively, resulting in a calculated $f\text{CO}_2$ of 356 µatm and a PA of 2317 µmol kg$^{-1}$ (Fig. 4, S1; Table S1). Measurements of carbonate chemistry during the acclimation period showed a stepwise increase in $f\text{CO}_2$, after which the $\text{CO}_2$ level remained largely

constant, with treatments ranging from 343 to 1641 µatm and a $pH_T$ range from 8.1 to 7.45 (Fig. 4; Table 1). Some decline in $f\text{CO}_2$ was observed in the high $\text{CO}_2$ treatments towards to the end of the experiment indicating that the addition of $\text{CO}_2$-saturated seawater was insufficient to fully compensate for its out-gassing into the headspace and drawdown by phytoplankton photosynthesis.

## 3.2 Nutrients

There was little variance in nutrient concentrations among all treatments at the start of the experiment (Table S1). Concentrations of NOx fell from $26.2 \pm 0.74$ uM on day 8 to below detection limits on day 18 (Fig. 5a), with the 1641 µatm treatment being drawn down the slowest. SRP concentrations were drawn down in a similar manner as NOx, falling from $1.74 \pm 0.02$ uM to $0.13 \pm 0.03$ uM on day 18 in all treatments (Fig. 5b). Silicate was replete throughout the experiment in all treatments, with initial concentrations of $60.0 \pm 0.91$ uM falling to $43.6 \pm 2.45$ uM (Fig. 5c). Silicate draw-down was highest in the 634

µatm and lowest in the 1641 µatm treatment.

## 3.3 Picophytoplankton abundance

Picophytoplankton abundance did not change during the $\text{CO}_2$ acclimation period and remained at $\sim 2.0 \pm 0.02$ x10$^6$ cells L$^{-1}$. Cell abundance increased in all treatments from day 8, with a significantly enhanced growth rate in the 953 µatm treatment when compared with the control (Table 2, 3). Abundance peaked on day 12 in treatments $\leq 506$ µatm at $5.5 \pm 0.61$ x10$^6$

cells L$^{-1}$ but continued to rise in treatments $\geq 634$ µatm until day 13 (Fig. 6a). Despite a faster growth rate in the 953 µatm treatment, peak abundance in this treatment was similar to the 1641 µatm treatment ($7.8 \pm 0.05$ x10$^6$ cells L$^{-1}$), while the 634 and 1140 µatm treatments peaked at a slightly lower abundance of $6.9 \pm 0.02$ x10$^6$ cells L$^{-1}$ (Fig. 6a). After reaching their peak, cell numbers rapidly declined in all treatments until day 18, falling to $0.8 \pm 0.03$ x10$^6$ cells L$^{-1}$. The 506 µatm treatment was excluded from analysis on day 18 due to very high background noise on the flow cytometer, resulting in artificially elevated

event counts.





Abundance curves for each $CO_2$ treatment were modelled from days 8 to 18, excluding the acclimation period when no growth occurred. The omnibus test of trends in picophytoplankton abundance among $CO_2$ treatments over time indicated there was no significant difference among treatments (Table 2, S2). However, examination of the model fits showed that whilst there was a reasonable fit to the data set (Adjusted $R^2$ = 0.82; Table 2), the constraints of limited data meant that the high abundance

values between days 12-14 in the treatments $\geq$634 µatm were not well fitted (Fig. S2). Despite this, the models did show the general trend of increased abundance in treatments $\geq$634 µatm. Analysis of the differences between peak abundances revealed that $CO_2$ treatments $\geq$634 µatm reached significantly higher maximum abundances than the control, while the 506 µatm treatment was significantly lower (Fig. 7a).

### 3.4    Nanophytoplankton abundance

Nanophytoplankton abundance declined during the $CO_2$ acclimation period in all treatments, falling from a mean initial abundance of $1.2 \pm 0.03$ x$10^6$ cells L$^{-1}$ to $0.9 \pm 0.02$ x$10^6$ cells L$^{-1}$ on day 7. Following acclimation, nanophytoplankton abundance increased in treatments $\leq$953 µatm until day 18, while treatments $\geq$1140 µatm remained low through to day 9 before increasing (Fig. 6b, S3). Analysis of steady-state logarithmic growth rates revealed that growth rates in the 634, 1140, and 1641 µatm treatments were significantly higher than the control (Table 2, 3). In spite of this, comparison of the trends between mod-

elled abundance curves for each $CO_2$ treatment indicated that the 634 and 953 µatm treatments were significantly enhanced compared to the control (Table 2, S3). In the 634 µatm $CO_2$ treatment, elevated nanophytoplankton abundance was observed from day 12 through to day 18, reaching a final abundance of $15 \pm 0.4$ x$10^6$ cells L$^{-1}$ (Fig. 6b). Despite lower abundance on days 8-9, enhanced growth rates in treatments $\geq$1140 µatm led to final abundances similar to the 953 µatm treatment on day 18, reaching $12 \pm 0.5$ x$10^6$ cells L$^{-1}$ (Fig. 6b, S3). The lowest nanophytoplankton abundance on day 18 was in the $CO_2$

treatments $\leq$506 µatm, which were $10 \pm 0.3$ x$10^6$ cells L$^{-1}$.

### 3.5    Heterotrophic nanoflagellate abundance

HNF abundance was initially low ($0.9 \pm 0.04$ x$10^5$ cells L$^{-1}$) and remained at a similar abundance throughout the $CO_2$ acclimation period. Abundance increased from day 8 in all treatments, but by day 9 was lower in $CO_2$ treatments $\geq$634 µatm than $\leq$506 µatm treatments, at $1.9 \pm 0.08$ x$10^5$ cells L$^{-1}$ and $2.9 \pm 0.18$ x$10^5$ cells L$^{-1}$, respectively and remained lower

until day 15 (Fig. 6c). Growth rate analysis between days 8 and 15 revealed that growth rates were significantly slower in the 506 µatm treatment and significantly faster in the 1641 µatm treatment, when compared with the control treatment (Table 2, 3). From day 15 to 18, the control, 634, and 953 µatm treatments continued to rise, reaching $3.2 \pm 0.07$ x$10^6$ cells L$^{-1}$, while abundance in the 506 µatm treatment stabilised between days 16 and 18, reaching $2.6 \pm 0.95$ x$10^6$ cells L$^{-1}$. HNF abundance remained lower than the control in the 1140 and 1641 µatm, reaching abundances on day 18 of $2.1 \pm 0.02$ x$10^6$ and $2.5 \pm$

$0.11$ x$10^6$ cells L$^{-1}$, respectively (Fig. 6c). The omnibus test among modelled abundance curves for each $CO_2$ treatment over time indicated that HNF abundance in at least one treatment differed significantly from the control (Table 2, S4). Examination of the significance of individual curve terms revealed that this reflected the significantly lower abundance of HNF in these two highest $CO_2$ treatments (1140 and 1641 µatm; Table 2).



### 3.6 Prokaryote abundance

Prokaryote abundance increased in $CO_2$ treatments $\geq$634 µatm during the acclimation period, with growth rates in treatments $\geq$953 µatm significantly higher than the control between days 4 and 8 (Table 2, 3). In contrast, abundance in treatments treatments $\leq$506 µatm remained unchanged (Fig. 6d). Between days 7 and 11, prokaryote abundance remained steady in all
treatments, with abundances in treatments $\geq$634 µatm significantly higher than the control (Fig. 7). During this time, the mean abundance was 3.09 $\pm$ 0.02 x$10^8$ cells L$^{-1}$ for treatments $\geq$953 µatm, 2.47 $\pm$ 0.02 x$10^8$ cells L$^{-1}$ in the 634 µatm treatment, and 2.07 $\pm$ 0.03 x$10^8$ cells L$^{-1}$ in treatments $\leq$506 µatm (Fig. 6d). After day 12, prokaryote abundance declined in all treatments, falling to 0.6 $\pm$ 0.06 x$10^7$ cells L$^{-1}$ by day 17.

Prokaryote abundance curves were modelled for each $CO_2$ treatment from days 4 to 18, excluding days 2 and 3 when no
growth occurred. There was no significant difference between $CO_2$ treatments in the omnibus test among modelled abundance curves (Table S5) but curves for the 953 and 1140 µatm treatments differed significantly from the control (Table 2). In a similar manner to the picophytoplankton data, the models did not well represent the high values in the treatments $\geq$953 µatm (Fig. S2). Whilst no significant differences were reported for the 634 and 1641 µatm treatments, the general trend in the modelled curves did follow that of the analysis, with increased abundance in all treatments $\geq$634 µatm.

### 3.7 Microbial community interaction

Although grazing experiments were not performed, the co-occurrence of slowed HNF growth with increased picophytoplankton and prokaryote abundance in $CO_2$ treatments $\geq$634 µatm suggests that the picophytoplankton and prokaryote communities were released from grazing pressure. Growth rates of prokaryotes and picophytoplankton were compared with HNF abundance on day 8 and 13, respectively, to examine whether trophic interactions could be inferred. Picophytoplankton had a negative but
non-significant trend (Fig. 8a; Table S6), while prokaryotes displayed a significant negative trend with HNF abundance (Fig. 8b; Table S7). This suggests that reduced HNF abundance reduced grazing mortality of the picoplankton community. This hypothesis was further supported by the observation that above a threshold HNF abundance there was a rapid decline in both the picophytoplankton and prokaryote abundance, irrespective of treatment and the duration of incubation. For picophytoplankton, this decline occurred when HNF abundance reached 0.84 $\pm$ 0.02 x$10^6$ cells L$^{-1}$ (Fig. 9a) and for prokaryotes it occurred after
HNF abundance reached 0.31 $\pm$ 0.02 x$10^6$ cells L$^{-1}$ (Fig. 9b). Interestingly, the decline in picophytoplankton and prokaryote abundances in the $CO_2$ treatments $\geq$634 µatm was greater than the control and 506 µatm treatments. However, this provided no benefit to HNF abundance in these treatments, which never surpassed that of the control (Fig. 6c).

## 4 Discussion

Mesocosm experiments are useful in assessing the effects of environmental perturbations on multiple trophic levels of a marine
ecosystem (Riebesell et al., 2008). Our results suggest that there are both direct effects of elevated $CO_2$ on nanophytoplankton



and indirect effects of trophic interactions occurring between HNF and their picoplanktonic prey that can significantly alter the composition and abundance of organisms at the base of the food web.

Exposing cells to a gradual change in $CO_2$ during an acclimation period allows cells an opportunity to adjust their physiology to environmental change and may alleviate some of the stress experienced when changes are imposed rapidly (Dason and
Colman, 2004). However, little is known about the time scales required for the changes in physiology necessary to optimise cellular tolerance of $CO_2$-induced stress. In addition, acclimating cells over the years to decades anticipated for anthropogenic ocean acidification is unachievable in most experimental designs. Acknowledging these limitations, a gradual increase in $fCO_2$ over 5 days was included in this study to assess whether acclimation would moderate the previously observed response of Antarctic microbial communities exposed to rapid changes in $CO_2$ (Davidson et al., 2016; Thomson et al., 2016; Westwood
et al., 2018).

The results of the current study were similar to those reported previously (Davidson et al., 2016; Thomson et al., 2016; Westwood et al., 2018) that lacked acclimation. Thus, it appears that an acclimation period had no discernible effect on the response of the community to enhanced $CO_2$. Hancock et al. (2018) did observe a significant change in microbial community composition in all treatments between days 1 and 3 but no further change in community composition was found between any
of the treatments during the acclimation. Therefore, they attributed this initial change to acclimation of the community to the minicosm tanks and not a response to increasing $CO_2$. This lack of acclimation may be due to ineffectiveness of the acclimation we used or to the highly variable $CO_2$ experienced by the marine microbial community at the study site. Here, $CO_2$ levels have been measured to vary by ~450 µatm throughout the year, with highest $CO_2$ levels experienced at the end of winter and strong $CO_2$ draw-down occurring in the Austral summer (Gibson and Trull, 1999; Roden et al., 2013). Marine organisms exposed to
highly variable environments have been shown to be more tolerant of changes in $CO_2$ (Boyd et al., 2016) and have also been demonstrated in this region (e.g Thomson et al., 2016; Deppeler et al., 2018).

It is also possible that the acclimation under low light conditions did not allow the cells to adjust their physiology effectively and that much of the acclimation occurred after the light levels were increased. Indeed, phytoplankton cell health (measured by photochemical quantum yield; $F_v/F_m$) was high during the low light acclimation period and a $CO_2$-induced decline in
health was only observed when light intensity was increased between days 5 and 8 (see Deppeler et al., 2018). Synergistic effects of $CO_2$ and light stress have been observed in a number of phytoplankton studies, with declines in growth, productivity, and cell health ($F_v/F_m$) reported under a combined high $CO_2$ and light intensity (Trimborn et al., 2017; Gao et al., 2012a, b; Li et al., 2015, e.g.). In our study, the phytoplankton community did appear to acclimate to this light and $CO_2$ stress, with $F_v/F_m$ increasing in all treatments after day 12 (Deppeler et al., 2018). Consequently, it is likely that the acclimation was either
incomplete or ineffective. Despite this, the similarity of our results with those previously reported does allow us to gain a more comprehensive understanding of the seasonal and temporal effects of ocean acidification on the marine microbial community in this region.



## 4.1 Heterotrophic nanoflagellates

Our study indicates that HNF abundance is negatively affected by elevated $CO_2$. This contrasts with the study by Moustaka-Gouni et al. (2016), who found no effect of $CO_2$ on the HNF community when exposed to levels up to 1040 ppm. As HNF cells are difficult to identify by microscopy in fixed samples (Sherr et al., 1993; Sherr and Sherr, 1993), we were unable to determine

whether the reduction in HNF abundance and differences in growth rates among treatments were due to $CO_2$-induced effects on the entire HNF community or if species-specific sensitivities changed the community composition. Hancock et al. (2018) reported a $CO_2$-related change in the relative abundances of two choanoflagellate species at $CO_2$ levels $\geq$634 $\mu$atm (see 4.4 below) and thus, it is possible that other $CO_2$-induced changes to HNF community composition may have occurred. Previous experiments in Prydz Bay, Antarctica also reported a reduction in HNF abundance when $CO_2$ was $\geq$750 $\mu$atm in both high

and low nutrient conditions (Thomson et al., 2016). The consistency of these results over the Austral summer and between years suggests that if $CO_2$ emissions continue to increase at rates similar to the IPCC RCP8.5 projections, the abundance and composition of HNF communities may change around 2050 (IPCC, 2013).

Increased top-down control by heterotrophic dinoflagellates and ciliates on the HNF community may have led to the lower abundance of HNF in the high $CO_2$ treatments. However, this was unlikely as Hancock et al. (2018) saw no effect of $CO_2$

on the composition or abundance of the microheterotrophic community in our study. Few other studies have investigated the effect of ocean acidification on heterotrophic protists and as yet there are no reports of direct effects of elevated $CO_2$ on microheterotrophic grazing rates, abundance, or taxonomic composition (Suffrian et al., 2008; Aberle et al., 2013). One study by Rose et al. (2009a) did report an increase in microzooplankton abundance when a natural North Atlantic microbial community was exposed to high $CO_2$ (690 ppm). However, this increased abundance was thought to be an indirect effect of

$CO_2$-induced promotion of phytoplankton abundance and a change in the phytoplankton community composition, as opposed to a direct effect of ocean acidification on microzooplankton physiology.

It is difficult to evaluate the potential reasons for reduced abundance in the HNF community in high $CO_2$ treatments as the mechanism(s) responsible for $CO_2$ sensitivity in HNFs are unstudied (Caron and Hutchins, 2013). Heterotrophs do not require $CO_2$ for growth, thus pH is likely the dominant driver of the effects observed (Sommer et al., 2015). The $CO_2$ sensitivity

of heterotrophic flagellates may be governed by the effectiveness of the mechanism(s) they possesses to regulate intracellular pH (Pörtner, 2008). However, little is known about the pH sensitivities of heterotrophic flagellates. Among the few studies on flagellates, a decline in pH influenced the swimming behaviour of a harmful algal bloom causing raphidophyte (Kim et al., 2013) and an inability to control intracellular pH disrupted the growth of the autotrophic dinoflagellates *Amphidinium carterae* and *Heterocapsa oceanica* (Dason and Colman, 2004). Disruption of flagella motility has also been observed in

marine invertebrate sperm, due to inhibition of the internal pH gradients required to activate signalling pathways (Nakajima, 2005; Morita et al., 2010; Nakamura and Morita, 2012). Whilst these examples do not provide evidence for direct inhibition of HNF growth, they do highlight the diverse sensitivities of flagellates to changes in pH that require further investigation. Size may also play a part in $CO_2$ sensitivity, with size-related declines in the external pH boundary layer meaning small cells are likely to be more affected by lower ocean pH (Flynn et al., 2012). As heterotrophs respire $CO_2$ and do not photosynthesise,





it is likely that pH would be even lower at the cell surface than for autotrophs. This may explain why HNFs showed reduced growth rates in our study while the larger microheterotrophs were unaffected (see Hancock et al., 2018).

This study highlights the need for additional research on the nanoflagellate community. There is an increasing understanding of the prevalence of mixotrophy in the marine microbial community (Gast et al., 2018; Mitra et al., 2014; Stoecker et al., 2017).

Mixotrophs are able to utilise both autotrophic and heterotrophic methods of energy production and consumption, although the methods employed can be diverse (Stoecker et al., 2017). It is currently unknown how mixotrophic phytoflagellates will respond to ocean acidification. Caron and Hutchins (2013) speculated that with an increasing concentration of DIC at increasing levels of $CO_2$, autotrophic energy production may be more efficient. However, the simultaneous increase in $H^+$ may have negative effects on both heterotrophic and autotrophic cellular mechanisms, causing multiple stresses to mixotrophic physiology. As

molecular methods are allowing for better identification of mixotrophic species (Gast et al., 2018), further research into how these species respond to increasing $CO_2$ may now be possible. Whilst iron was not a limiting factor for phytoplankton in the coastal region studied (Davidson et al., 2016), it is a significant driver on the ecology of the marine microbial community in a majority of the Southern Ocean (Martin et al., 1990). Iron limitation has been found to lessen the impact of $CO_2$ on some diatom species, especially in combination with other stressors (Hoppe et al., 2013). No studies to date have investigated the

effect of ocean acidification on HNF in the iron-limited Southern Ocean, despite their dominance in the microbial community this region (Safi et al., 2007). Thus, it is imperative that further study be done.

## 4.2   Nano- and picophytoplankton

A significant increase in picophytoplankton abundance was observed in our study when $CO_2$ levels were $\geq 634$ µatm (Fig. 6a). Increased abundance of picophytoplankton has been reported in ocean acidification studies on natural communities around

the world (e.g. Brussaard et al., 2013; Schulz et al., 2013; Biswas et al., 2015; Crawfurd et al., 2017). In contrast, Antarctic community studies report varying responses to elevated $CO_2$. Shifts toward larger diatom species have been reported in coastal waters of the Ross Sea (Feng et al., 2010; Tortell et al., 2008), while there was no $CO_2$-induced change to growth or community composition at a site on the Antarctic Peninsula (Young et al., 2015). This variability in response among sites in Antarctic waters may be due to factors such as differences in microbial composition or study methods. Picophytoplankton were either

not counted (Feng et al., 2010; Tortell et al., 2008) or were considered negligible (Young et al., 2015) in these studies. The significant increase in picophytoplankton abundance at $CO_2$ levels $\geq 634$ µatm that we report is similar to the findings of Thomson et al. (2016) at the same site and using similar methods, indicating that this response is consistent across different seasonal and temporal environments. It has been suggested that increased abundance of picophytoplankton may be due to increases in productivity derived from more readily-available $CO_2$ at the cell surface, allowing more passive diffusion of $CO_2$

into the cell, and thus, reduced requirements for energy-intensive carbon concentration mechanisms (CCMs) (Riebesell et al., 1993; Paulino et al., 2008; Schulz et al., 2013; Calbet et al., 2014). CCMs were down-regulated in the high $CO_2$ (1641 µatm) treatment in both small (<10 µm) and large ($\geq 10$ µm) cells in our study (Deppeler et al., 2018). We did not observe any increase in primary productivity from CCM down-regulation in this treatment (Deppeler et al., 2018) although, small changes





in exponential growth get amplified over time and are difficult to pick up in primary productivity measurements, which are representative for the entire community.

Larger cell surface to volume ratios in small cells, allowing increased nutrient utilisation in nutrient-limited environments, has also been invoked to explain the increased abundance of picophytoplankton with elevated $CO_2$ (Schulz et al., 2013). Size-related differences in growth rates may allow picophytoplankton to establish a bloom faster than larger phytoplankton species (e.g. Newbold et al., 2012). However, this is not seen in nutrient-replete Antarctic waters, where early summer blooms are dominated by large diatoms and *Phaeocystis antarctica* in its colonial life-stage (Davidson et al., 2010). It was also not observed in this study, where only the 953 µatm treatment displayed a significantly enhanced growth rate (Table 2). Increased rates of nutrient draw-down were observed in the 634-953 µatm $CO_2$ treatments (Fig. 5), suggesting that moderate increases in $CO_2$ may stimulate phytoplankton growth, but further increases in $CO_2$ led to significant reductions in primary productivity (Deppeler et al., 2018).

Nanophytoplankton abundance was significantly higher in the 643 and 953 µatm treatments, with significantly increased growth rates in the 634, 1140, and 1641 µatm treatments (Fig. 6b; Table 2). This was likely due to favourable conditions, including the inhibition of growth of larger phytoplankton species, that allowed nano-sized phytoplankton to thrive at higher $CO_2$ levels (Hancock et al., 2018). The initial decline in nanophytoplankton abundance in all treatments between days 1 and 7 may have been due to acclimation of the community to the mesocosms or grazing by microzooplankton. Increasing light intensity had a temporary inhibitory effect on growth at $CO_2$ levels $\geq$1140 µatm between days 8 and 9 (Fig. 6b), suggesting that the significantly enhanced growth rates in these treatments between days 9 and 15 may have been caused by an increase in relative abundance of more tolerant species. The most abundant nanophytoplankton species present in the minicosms were *Fragilariopsis* spp. and *Phaeocystis antarctica* in it's colonial form (Hancock et al., 2018). These species displayed a $CO_2$-related threshold in dominance around 634 µatm, with a shift from *P. antarctica* to *Fragilariopsis* spp. in the high $CO_2$ treatments (Hancock et al., 2018). Thus, it is likely that relative fitness of both of these species is increased with a moderate increase in $CO_2$ level, explaining the higher abundance observed at 643 and 953 µatm $CO_2$. Interestingly, whilst no negative effect of $CO_2$ was observed on the overall nanophytoplankton abundance, there were very strong species-specific responses to increasing $CO_2$, resulting in a significant change in community structure (Hancock et al., 2018). Increased abundance of *Fragilariopsis* spp. with elevated $CO_2$ has also been observed in other ocean acidification studies on natural Antarctic microbial communities (Hoppe et al., 2013; Davidson et al., 2016). Therefore, it is likely that increasing $CO_2$ will cause the phytoplankton community to shift from a summer community that is currently dominated by large diatoms to one composed of smaller species or morphotypes of nano- and picophytoplankton.

## 4.3 Prokaryotes

There was a significant increase in abundance of prokaryotes at $CO_2$ levels $\geq$634 µatm (Fig. 6d; Table 2). Increases prokaryote abundance with elevated $CO_2$ was also observed in previous studies at Prydz Bay (Thomson et al., 2016), as well as in Arctic mesocosms (Endres et al., 2014; Engel et al., 2014). Other studies have reported no influence of $CO_2$ on the prokaryote community (Grossart et al., 2006; Allgaier et al., 2008; Paulino et al., 2008; Newbold et al., 2012), suggesting that the prokaryote





community will tolerate increasing $CO_2$ levels (Reviewed in Hutchins and Fu, 2017). Like HNF, prokaryotes do not require $CO_2$ for growth, although it appears they are more resistant to large variations in pH. However, there is evidence that $CO_2$ may affect prokaryotes by inducing changes in community composition, selecting for more tolerant species or allowing rare species to emerge (Krause et al., 2012; Roy et al., 2013; Zhang et al., 2013; Bergen et al., 2016). This may be related to differential re-
sponses of phylogenetic groups to maintaining pH homeostasis in either acid and alkaline conditions (Padan et al., 2005; Bunse et al., 2016). The mechanisms for transporting hydrogen ions ($H^+$) out of the cell are energetically demanding and may reduce the energy available for growth. Whether these energy demands are increased or decreased with ocean acidification depends upon the different strategies for pH homeostasis employed by individual prokaryote species (Teira et al., 2012). In their study, Teira et al. (2012) observed a significant increase in growth efficiency with elevated $CO_2$ in one bacterial strain, although
no increase in productivity or abundance resulted. Instead, these changes may affect dissolved organic carbon consumption (Endres et al., 2014), with potential impacts on organic matter cycles.

### 4.4 Community interactions

The coincidence of the increase in picophytoplankton and prokaryote abundances with reduced abundance of HNF suggests that these communities were being released from grazing pressure at $CO_2$ levels $\geq$634 µatm. Grazing rates in East Antarctica
are on average, 62% of primary production per day, up to a maximum of 220% (Pearce et al., 2010). In addition, >100% of prokaryote production can be removed by micro- and nanoheterotrophs when Chl $a$ concentration and prokaryote abundance is high (Pearce et al., 2010). The rapid decline in abundance we observed in picophytoplankton and prokaryotes after 12 days incubation is entirely consistent with the rapid rates of grazing observed in other Antarctic marine microbial communities in this region. In relation to $f\mathrm{CO}_2$, it is reasonable to hypothesise that the lower abundances of these prey sizes in the control and
506 µatm treatments may have been due to stronger top-down control on the community as opposed to a reduction in growth rate. Grazing control of the picophytoplankton community has been proposed in other mesocosm studies to explain both positive (Paulino et al., 2008; Rose et al., 2009a) and negative (Meakin and Wyman, 2011; Newbold et al., 2012) changes in picophytoplankton abundance, although they were not confirmed by HNF counts. In our study, the rapid decline in prokaryote abundance coincided with a dramatic increase in choanoflagellate abundance, bactivorous eukaryotes, between days 14 and
16 (Hancock et al., 2018). Furthermore, picophytoplankton and prokaryotes in all $CO_2$ treatments both declined after HNF abundance reached a critical threshold (Fig. 9), suggesting that at this point their growth was unable to exceed the top-down control of grazing.

Species-specific differences in the sensitivity of HNF to $CO_2$ may lead to significant changes in the composition of the picophytoplankton and prokaryote communities. HNF food webs are complex and successional changes in taxa occur during
phytoplankton blooms (Moustaka-Gouni et al., 2016). In our study, Hancock et al. (2018) observed species-specific differences in the $CO_2$ tolerances of choanoflagellate species, where *Bicosta antennigera* displayed significant $CO_2$ sensitivity at levels $\geq$634 µatm while other choanoflagellate species (principally *Diaphanoeca multiannulata*) were unaffected. This change in HNF community composition with increased $CO_2$ did not affect the total prokaryote abundance but may have implications for the prokaryotic community composition through selective grazing. Changes in prokaryote community composition have



been observed in other mesocosm studies (Roy et al., 2013; Zhang et al., 2013; Bergen et al., 2016). There is also evidence that different prokaryote phylogenetic groups have preferences for organic substrates produced by different phytoplankton taxa (Sarmento and Gasol, 2012), leading to the possibility that future changes in prokaryote community composition could impact organic matter recycling.

As viral abundance was not determined in our study, we cannot exclude viral lysis as an explanation for the rapid decline in picophytoplankton and prokaryote abundance. Viral lysis can account to up to 25% of daily production, although grazing by micro- and nanoheterotrophs can be twice as high (Evans et al., 2003; Pearce et al., 2010). In an Arctic mesocosm study, the decline of a picophytoplankton bloom coincided with a large increase in viral abundance (Brussaard et al., 2013). However, later in the study, picophytoplankton were heavily grazed by microzooplankton. Bacteriophages are the dominant viruses in the

Prydz Bay area (Pearce et al., 2007; Thomson et al., 2010; Liang et al., 2016), with viral abundance displaying no correlation to picophytoplankton (Liang et al., 2016). This suggests that viral lysis was unlikely to be the main cause of the decline in picophytoplankton numbers but may have affected the prokaryotes.

## 5    Conclusions

The results of this study show how ocean acidification can exert both direct and indirect influences on the interactions among

trophic levels within the microbial loop. Our study reinforces findings in near shore waters off East Antarctica (Davidson et al., 2016; Thomson et al., 2016) that HNF abundance is reduced when $CO_2$ is ≥634 μatm, irrespective of temporal changes in the physical and biological environment among seasons and years. This likely resulted in a decline in grazing mortality of picophytoplankton and prokaryotes, allowing these communities to increase in abundance. Such changes in predator-prey interactions with ocean acidification could have significant effects on the food web and biogeochemistry in the Southern

Ocean. HNF are an important link in carbon transfer to higher trophic levels as they are grazed upon by microzooplankton and thereafter by higher trophic organisms (Azam et al., 1991; Sherr and Sherr, 2002). Grazing is also a critical determinant of phytoplankton community composition and standing stocks (Sherr and Sherr, 2002).

   Our results, together with those of Deppeler et al. (2018) and Hancock et al. (2018), indicate it is likely that increasing $CO_2$ will cause a shift away from blooms dominated by large diatoms towards communities increasingly dominated by prokaryotes,

nano- and picophytoplankton. Large phytoplankton cells contribute significantly to deep ocean carbon sequestration (Tréguer et al., 2018). They are also the preferred food source for higher trophic organisms, especially the Antarctic krill *Euphausia superba* (Haberman et al., 2003; Meyer et al., 2003; Schmidt et al., 2006). *E. superba* have been found to graze less efficiently on phytoplankton cells <10 μm (Quetin and Ross, 1985; Kawaguchi et al., 1999; Haberman et al., 2003). Therefore, a shift to smaller-celled communities will likely alter the structure of the Antarctic food web. Furthermore, increases in prokaryote

abundance will likely intensify the breakdown of organic matter in surface waters, further contributing in a decline in the sequestration of carbon from summer phytoplankton blooms into the deep ocean.



*Data availability.* Experimental data used for analysis are available via the Australian Antarctic Data Centre.

Environmental data: Deppeler, S.L., Davidson, A.T., Schulz, K.: Environmental data for Davis 14/15 ocean acidification experiment, Australian Antarctic Data Centre, http://dx.doi.org/10.4225/15/599a7dfe9470a, 2017, (updated 2017).

Flow cytometry data: Deppeler, S.L., Schulz, K.G., Hancock, A., Pascoe, P., Mckinlay, J., Davidson, A.T. (2018, updated 2018) Data for

5   manuscript 'Ocean acidification reduces growth and grazing of Antarctic heterotrophic nanoflagellates' Australian Antarctic Data Centre, http://dx.doi.org/10.4225/15/5b234e4bb9313, 2018 (updated 2018)

Micoscopy data: Hancock, A.M., Davidson, A.T., Mckinlay, J., Mcminn, A., Schulz, K., Van Den Enden, D. (2017, updated 2018) Ocean acidification changes the structure of an Antarctic coastal protistan community Australian Antarctic Data Centre, http://dx.doi.org/10.4225/ 15/592b83a5c7506, 2018 (updated 2018)

10   *Author contributions.* AD conceived and designed the experiments. AD led and oversaw the minicosm experiment and PP, SD, and AH performed the experiments. SD and AD performed the data analysis. KS performed the carbonate system measurements and manipulation. JM provided statistical guidance. SD wrote the manuscript with all other authors providing contributions and critical review of the manuscript.

*Competing interests.* The authors declare that they have no conflict of interest.

*Acknowledgements.* This study was funded by the Australian Government, Department of Environment and Energy as part of Australian

15   Antarctic Science Project 4026 at the Australian Antarctic Division and an Elite Research Scholarship awarded by the Institute for Marine and Antarctic Studies, University of Tasmania. We would like to thank Prof. Dave Hutchins and Prof. Scarlett Trimborn for valuable comments on this manuscript. We gratefully acknowledge the assistance of AAD technical support in designing and equipping the minicosms and Davis Station expeditioners in the summer of 2014/15 for their support and assistance.



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

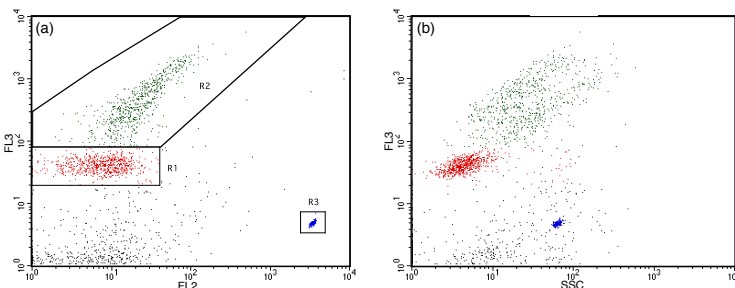

**Figure 1.** Nano- and picophytoplankton regions identified by flow cytometry. (a) Two separate regions identified based on red (FL3) versus orange (FL2) fluorescence scatter plot. (b) Picophytoplankton (R1) and nanophytoplankton (R2) communities determined from side scatter (SSC) versus FL3 fluorescence scatter plot. PeakFlow Green 2.5 µm beads (R3) used as fluorescence and size standard.

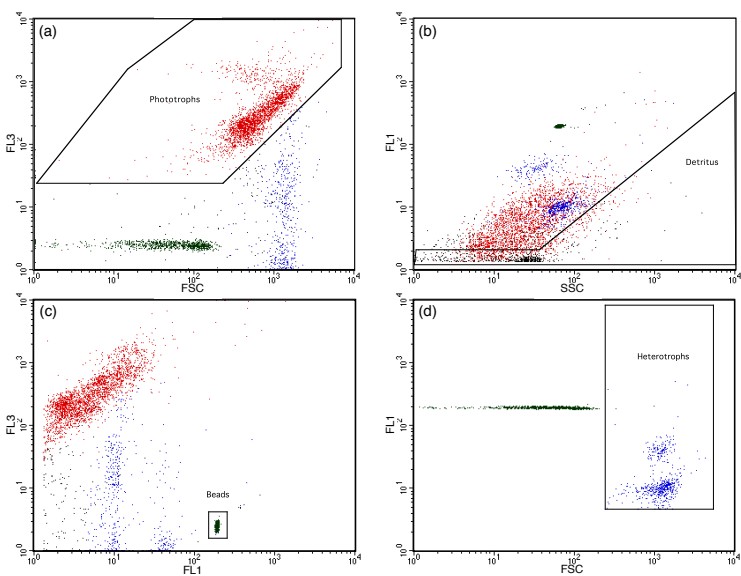

**Figure 2.** LysoTracker Green-stained heterotrophic nanoflagellates identified by flow cytometry. (a) Phytoplankton identified based on red (FL3) versus orange (FL2) fluorescence scatter plots. (b) Detritus particles identified from high side scatter (SSC) versus LysoTracker Green fluorescence (FL1). (c) PeakFlow Green 2.5 µm beads identified from high FL1 versus low red (FL3) fluorescence. (d) Phytoplankton and detritus from (a) and (b) removed from FL1 and forward scatter (FSC) plot and remaining LysoTracker Green-stained particles >2.5 µm were counted as heterotrophic nanoflagellates.





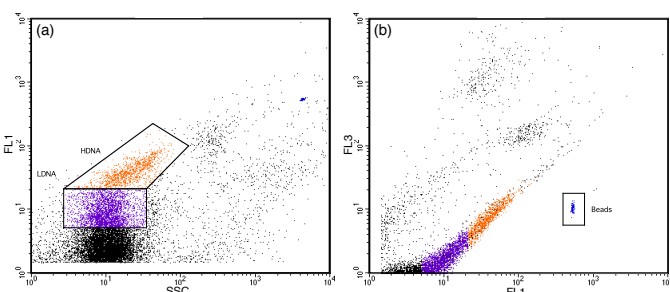

**Figure 3.** Prokaryote regions identified by flow cytometry. (a) SYBR-Green I-stained high DNA (HDNA) and low DNA (LDNA) prokaryote regions identified from side scatter (SSC) versus green fluorescence (FL1) scatter plots. (b) Prokaryote cells determined from high FL1 versus low red (FL3) fluorescence. PeakFlow Green 2.5 μm beads used as fluorescence and size standard.

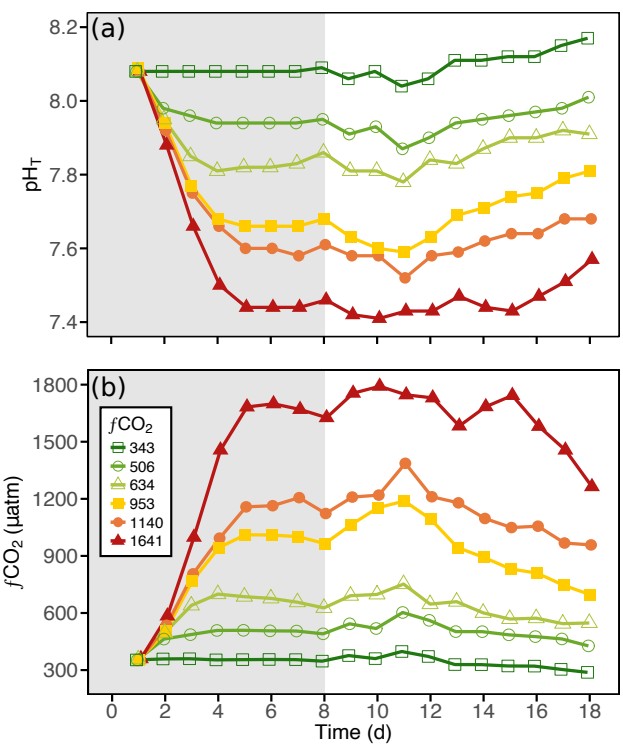

**Figure 4.** The (a) pH on the total scale ($pH_T$) and (b) fugacity of $CO_2$ ($fCO_2$) carbonate chemistry conditions in each of the minicosm treatments over time. Grey shading indicates $CO_2$ and light acclimation period.

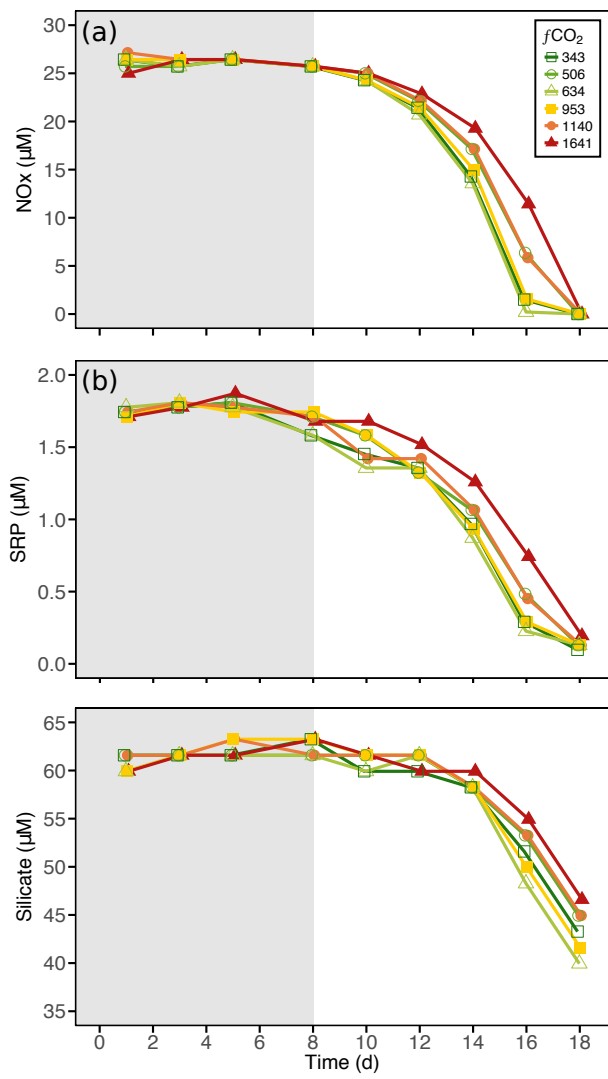

**Figure 5.** Nutrient concentration in each of the minicosm treatments over time. (a) Nitrate + nitrite (NOx), (b) soluble reactive phosphorus (SRP), and (c) molybdate reactive silica (Silicate). Grey shading indicates $CO_2$ and light acclimation period.



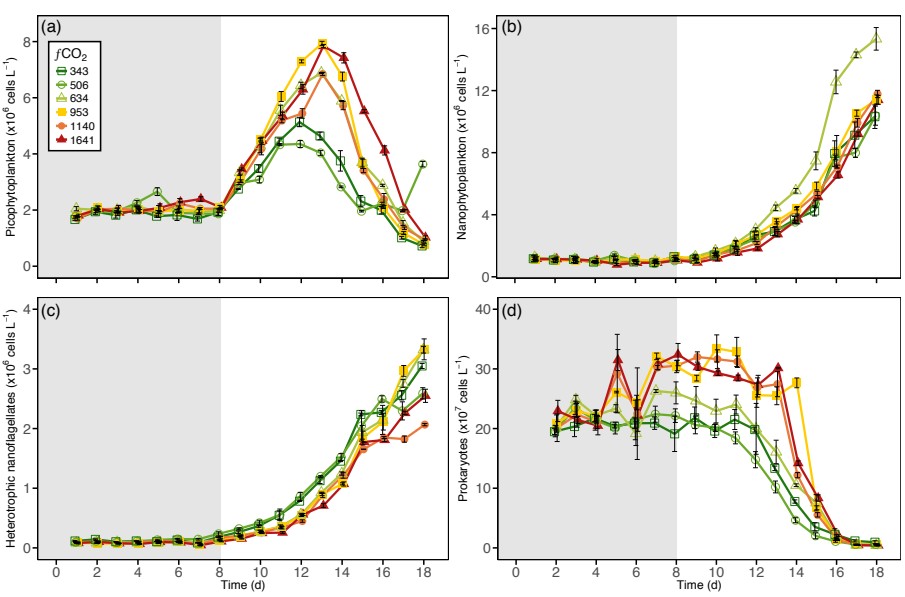

**Figure 6.** Abundance of (a) picophytoplankton, (b) nanophytoplankton, (c) heterotrophic nanoflagellates, and (d) prokaryotes in each of the minicosm treatments over time. Error bars display standard error of pseudoreplicate samples. Grey shading indicates $CO_2$ and light acclimation period.

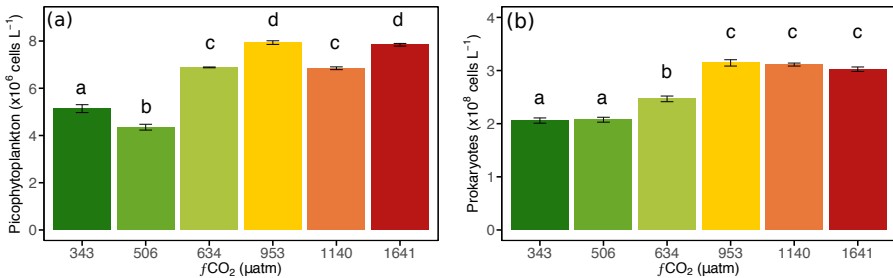

**Figure 7.** Peak abundances of (a) picophytoplankton and (b) prokryotes in each of the minicosm treatments. Letters indicate significantly different groupings assigned by post-hoc Tukey test. Error bars display standard error of pseudoreplicate samples.





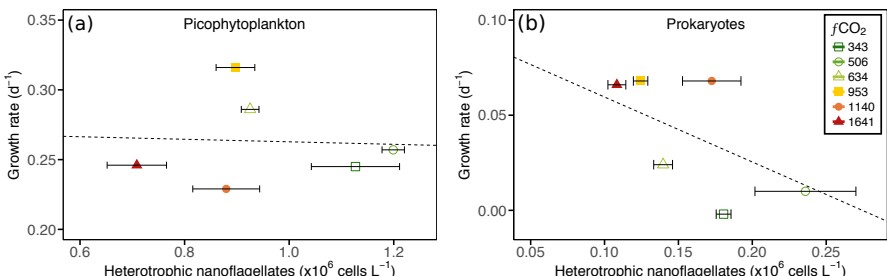

**Figure 8.** Comparison of (a) picophytoplankton (day 13) and (b) prokaryote (day 8) steady-state growth rates against heterotrophic nanoflag-ellate abundance. Error bars display standard error of pseudoreplicate samples of heterotrophic nanoflagellates. Dotted line indicates linear regression trend (Data in Table S6, S7).

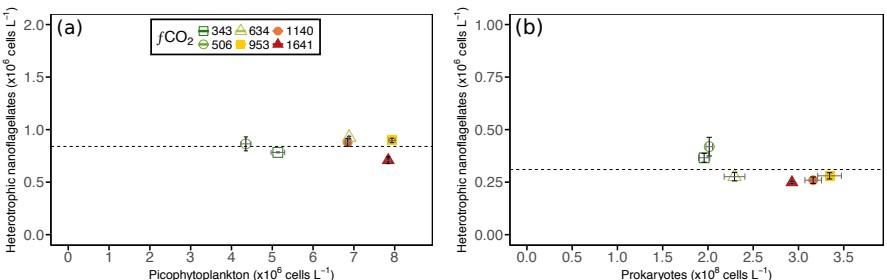

**Figure 9.** Heterotrophic nanoflagellate abundance on the day before (a) picophytoplankton and (b) prokaryote abundance declined in each of the minicosm treatments. Error bars display standard error of pseudoreplicate samples of heterotrophic nanoflagellates (grey) and picophytoplankton/prokaryotes (black). Dotted line indicates threshold of heterotrophic nanoflagellate abundance of (a) $0.84 \pm 0.02$ x$10^6$ cells L$^{-1}$ and (b) $0.31 \pm 0.02$ x$10^6$ cells L$^{-1}$.



**Table 1.** Mean carbonate chemistry conditions in minicosms

| Tank | $f\mathrm{CO}_2$ (µatm) | $\mathrm{pH}_T$ | DIC (µmol kg$^{-1}$) | PA (µmol kg$^{-1}$) |
|---|---|---|---|---|
| 1 | $343 \pm 30$ | $8.10 \pm 0.04$ | $2188 \pm 6$ | $2324 \pm 11$ |
| 2 | $506 \pm 43$ | $7.94 \pm 0.03$ | $2243 \pm 8$ | $2325 \pm 10$ |
| 3 | $634 \pm 63$ | $7.85 \pm 0.04$ | $2270 \pm 5$ | $2325 \pm 12$ |
| 4 | $953 \pm 148$ | $7.69 \pm 0.07$ | $2314 \pm 11$ | $2321 \pm 11$ |
| 5 | $1140 \pm 112$ | $7.61 \pm 0.04$ | $2337 \pm 5$ | $2320 \pm 10$ |
| 6 | $1641 \pm 140$ | $7.45 \pm 0.04$ | $2377 \pm 8$ | $2312 \pm 10$ |

Data are mean $\pm$ one standard deviation of triplicate pseudoreplicate measurements



**Table 2.** ANOVA results comparing trends in each $CO_2$ treatment over time against the control

|  | F | Adjusted $R^2$ | Day:506 p-value | Day:634 p-value | Day:953 p-value | Day:1140 p-value | Day:1641 p-value |
|---|---|---|---|---|---|---|---|
| *Modelled growth curves* | | | | | | | |
| Pico | $F_{12,182} = 74.6$ | 0.82 | 0.38 | 0.80 | 0.57 | 0.76 | 0.08 |
| Nano | $F_{12,311} = 478.8$ | 0.95 | 0.47 | **<0.01** | **0.01** | 0.10 | 0.78 |
| HNF | $F_{12,307} = 634.3$ | 0.96 | 0.15 | 0.88 | 0.99 | **<0.01** | **<0.01** |
| Prok | $F_{12,256} = 131.5$ | 0.85 | 0.39 | 0.49 | **<0.05** | **0.04** | 0.08 |
| *Steady-state growth rate* | | | | | | | |
| Pico | $F_{11,81} = 144.7$ | 0.95 | 0.71 | 0.12 | **<0.01** | 0.48 | 0.98 |
| Nano | $F_{11,132} = 611.1$ | 0.98 | 0.34 | **<0.01** | 0.29 | **<0.05** | **0.01** |
| HNF | $F_{11,131} = 518.6$ | 0.98 | **0.02** | 0.30 | 0.32 | 0.39 | **0.02** |
| Prok | $F_{11,113} = 12.94$ | 0.51 | 0.52 | 0.17 | **<0.01** | **<0.01** | **<0.01** |

Bold text denotes significant p-values (<0.05). Pico; picophytoplankton, Nano; nanophytoplankton, HNF; heterotrophic nanoflagellates, Prok; prokaryotes.



**Table 3.** Steady-state logarithmic growth rates in $CO_2$ treatments

|      | 343 µatm | 506 µatm | 634 µatm | 953 µatm | 1140 µatm | 1641 µatm |
|------|------|------|------|------|------|------|
| Pico | 0.25 | 0.26 | 0.29 | **0.32** | 0.23 | 0.25 |
| Nano | 0.26 | 0.25 | **0.32** | 0.27 | **0.28** | **0.29** |
| HNF  | 0.36 | **0.32** | 0.38 | 0.37 | 0.34 | **0.40** |
| Prok | 0.00 | 0.01 | 0.02 | **0.07** | **0.07** | **0.07** |

Bold text denotes growth rates significantly different to the control (343 µatm, $p < 0.05$). Pico; picophytoplankton, Nano; nanophytoplankton, HNF; heterotrophic nanoflagellates, Prok; prokaryotes.