# Peer review of "Ocean acidification reduces growth and grazing impact of Antarctic heterotrophic nanoflagellates"

_Biogeosciences, 2019_

## Referee Comment (RC1) · Anonymous Referee #1 · 17 Jul 2019

General comments This work is part of a minicosm investigation of the effects of increasing fCO2 levels on a natural planktonic microbial community of Prydz Bay, East Antarctica, and deals with the response of heterotrophic flagellates (HNF), nano- and picophytoplankton, and prokaryotes. The design of the experiments was similar to that of previous studies in East Antarctica, but with an initial CO2 acclimation period. The present manuscript complements other publications (one of them, at least, in Biogeosciences) on the same minicosm experiment, and will have benefitted from the reviews of the previous works. Overall, the manipulations appear to have been competently carried out and the text is well written. Concerning the discussion, I appreciated, in particular, the consideration given to potential community shifts, in addition to physio-

logical changes. Some comments on aspects that could be improved are given below. Specific comments The main results of these accompanying works tend to appear late in the text; they should rather be presented up front in the introduction, so that the reader can better appreciate what is the context for and the contribution of the present study. Some conclusions go further than supported by the presented results. For example, the statement (whether correct or not) : "Therefore, it is likely that increasing CO2 will cause the phytoplankton community to shift from a summer community that is currently dominated by large diatoms to one composed of smaller species or morphotypes of nano- and picophytoplankton." (lines 27-29 of page 13) does not derive from the work shown in the present manuscript (or if the authors believe so, it should be much better discussed). Other comments It would be helpful for the readers to give more details on the statistical analyses (for example, explain "I" in tables S2-S5, number of time points and of pseudoreplicates). It would be helpful to repeat somewhere that the prokaryote group here is supposed to include few or no cyanobacteria. Line 3 of page 9. Eliminate "treatments". Lines 6-7 of page 10. "acclimating cells over the years to decades . . . is unachievable in most experimental designs". It is also doubtful to expect that the same cells/taxa would be acclimating for years or decades in natural settings. Lines 7-8 of page 13. "dominated by large diatoms and . . ." Which were the main "large diatom taxa"? Lines 28-29 of page 13. "a summer community that is currently dominated by large diatoms". This would not apply to many Antarctic areas. Line 31 of page 13. "Increases of prokaryote .." Explanation of Fig. 7: "prokaryotes" instead of "prokryotes". Explanation of Fig. 9: Add indication that the abscisssa shows the picophytoplankton and prokaryote abundances on the day before decline. For example: "Heterotrophic nanoflagellate abundance (y axis) on the day before (a) picophytoplankton and (b) prokaryote abundance (shown in x axis) declined in each . . . "

---

## Referee Comment (RC2) · Anonymous Referee #2 · 15 Mar 2020

General Comments

Heterotrophic nanoflagellates play and important role in pelagic marine ecosystems as grazers of picoplankton and as prey for microheterotrophs (ciliates and heterotrophic dinoflagellates). Polar marine ecosytems are especially vulnerable to ocean acidification and a several studies have investigated the effects of Ocean acidification on Antarctic and Arctic pelagic microbial communities. This paper reports the results of a study which employed experimental minicosms to examine the effects of ocean acidification on pelagic microbial communities in Antarctica coastal waters. It contributes new observations on predator-prey interactions in response to acidification and sup-

ports observations derived from a previous study undertaken the same location using a similar experimental approach. The novelty of the study, as compared to the previous study, lies in the incorporation of an initial acclimation period within the experimental design. It is also useful and somewhat novel to encounter a paper which repeats and reinforce the insights gained from earlier work. The paper is well presented with a well-referenced introduction and discussion. The methods and data interpretation are sound, and the conclusions are supported by the results. However, there are some weaknesses in data interpretation and conclusion, outlined below, which should be addressed. In summary, the paper should make a valuable contribution to the literature addressing a timely and relevant topic which should be of interest to readers of Biogeochemistry.

Specific Comments

Section 2.4 (Page 5, line 19): State volume of the single sample removed from each minicosm on each day for flow cytometry analysis (from which different sets of pseudoreplicates were subsequently removed for analysis of different microbial groups).

Section 2.4 (Page 5, line 22): Flow cytometry is the main method used to generate microbial data in the study so the authors should describe in full how flow cytometer sample flow rates were calibrated. This is important as flow cytometer flow rates are highly variable, and the exact volume analysed from each sample must be assessed independently. Poor calibration technique is therefore a significant source of error in some studies. Describing how flow rates were calibrated gives confidence that the microbial abundance data are accurate. It also reminds readers that rigorous procedures are required to generate accurate data from flow cytometers.

Section 2.4.3 (Page 6, line 18): The prokaryote abundance measurements (undertaken by flow cytometry using SYBR Green I stain and FL1 versus SSC plots) will include phototrophic prokaryotes (i.e. picophytoplankton) as well as heterotrophic prokaryotes, unless the picophytoplankton data (derived from analyses in Section 2.4.1) were

subtracted from the prokaryote counts. This should be stated in this section of the methods. Heterotrophic prokaryotes will dominate these data, especially in Antarctic waters, so it is acceptable to treat the prokaryote results as representing mainly heterotrophic bacteria in subsequent discussions.

Section 2.5 (Page 7, Line 3): The authors correctly state that "...statistical differences among treatments should be interpreted conservatively..." due to lack of true replication. Clear trends between treatments can be clearly identified over the duration of incubation. However, conclusions based on the analysis of statistically significant differences between treatments (based on pseudoreplicates) at any one time point (page 7, line 1) are unconvincing. These include the subsequent statements (page 8, line 7; page 9, line 5) based on the picophytoplankton and prokaryote peak abundance analyses shown in Figure 7. Also the comparison of picophytoplankton and prokaryote growth rates with heterotrophic nanoflagellate abundance (page 9, line 20) shown in Figure 8. The respective conclusions from these analyses (that picoplankton and prokaryote abundance differ between treatments, and that reduced heterotrophic nanoflagellate abundance reduced grazing on picoplankton) are well-supported by the other analyses using data from several time points. I therefore question whether the authors should include the analyses presented in Figures 7 and 8.

Sections 3.4 (Page 8, line 15) and Section 4.2 (page 13, lines 12 and 23): I see no evidence in Fig 6b or Fig S3b that nanophytoplankton abundance was higher than the control in the 954 uatm treatment. The modelling data (shown in Table 2) may have revealed this but the modelling result is a simulation of the underlying data which, in turn, is based on pseudoreplicates. The figures clearly show that nanophytoplankton abundance was higher than the control in the 634 treatment, but the case for the 954 uatm treatment resulting in higher abundance is unconvincing.

Section 4.1 (Page 11, line 10): I am not convinced of the utility of the conclusion that heterotrophic nanoflagellate communities may change by 2050 due to ocean acidification. The abundance and composition of Antarctic heterotrophic nanoflagellate

communities may well change by 2050 for many reasons, and microcosm experiments undertaken over 18 days cannot simulate real environmental changes to entire ecosystems over decades. I suggest this conclusion is removed.

Section 4.1 (Page 11, line 13): The discussion on top-down control of heterotrophic nanoflagellates by the microheterotrophic community (heterotrophic dinoflagellates and ciliates) could include some additional considerations, as follows:

First, the study by Hancock et al. (2018) assessed microheterotroph abundance in Lugol's fixed samples of 2 to 10ml volume. It is not possible to derive meaningful microheterotroph data from such small sample volumes, so the statement (page 11, line 14) that treatments had no effect on the heterotrophic dinoflagellates and ciliates may not be valid.

Second, it would be useful to discuss the evidence for any switching in grazing pressure by microheterotrophs between nanophytoplankton and heterotrophic nanoflagellates. The fact that nanophytoplankton abundance was similar between treatments (except for 634 uatm – Figure 6b) suggests that heterotrophic dinoflagellates and ciliates were not exhibiting differential grazing pressure on heterotrophic nanoflagellates between treatments. This, in turn, lends support to the conclusion that the lower heterotrophic nanoflagellate abundances in high $CO_2$ treatments were not a result of top-down pressure (assuming microheterotroph numbers were not affected by acidification and similar in each treatment). On the other hand, the observed shift in the composition of the nanophytoplankton community from Phaeocystis to Fragilariopsis in high $CO_2$ treatments (page 13, line 21), as reported by Hancock et al. (2018), suggests that one would expect some differential microheterotrophic grazing between treatments and possible switching between nanophytoplankton and heterotrophic nanoflagellate prey.

Third, the consequences of screening the seawater used to fill the minicosm tanks through a 200 micron filter should be discussed. This action will have reduced top-down grazing pressure on microheterotrophs, possibly creating a differential trophic

cascade effect between treatments over the 18 days incubation. Any such effects may well have been minimal and equal across treatments. However, the potential effect of initial seawater screening should be discussed, especially with respect to the limits to which minicosm experiments can simulate the dynamics of in situ communities.

Section 4.1 (Page 12, line 5): Mixotrophic nanoflagellates will have been included within the nanophytoplankton counts due to the presence of chlorophyll (albeit possibly at low levels) within the cells. This should be mentioned in the methods or discussion text.

Section 4.3 (Page 13, line 32) and Section 4.4 (page 14, line 34): The results of Westwood et al. (2018) should be discussed as they are derived from the same location and draw similar conclusions to the present study (i.e. enhanced bacterial production and abundance in high $CO_2$ treatments coinciding with reduced heterotrophic nanoflagellate abundance).

Technical Corrections

Page 1, Title: The title could be misinterpreted as reporting the effects of ocean acidification on the "...grazing of heterotrophic nanoflagellates." by their microzooplankton predators. A more accurate but unwieldly wording would be "...reduces growth of and grazing on heterotrophic nanoflagellates.". Perhaps rephrase as "...reduces growth and grazing impact of heterotrophic nanoflagellates".

Page 2, line 6: Correct spelling "whish" to "which".

Page 2, line 27: Use of the phrase "...in the present study..." implies that the observations referred to are part of the submitted manuscript rather than a different publication. Perhaps use the phrase "...concurrently observed amongst choanoflagellates in the present minicosm experiment..."?

Page 6, line12: The text refers to Figure 2a which shows a plot of FL3 versus FSC, rather than FL3 versus FL2 as stated in text.

Page 9, Line 5: "Fig. 7" should read "Fig. 7b".

Page 11, line 27: Add hyphen to change text to "...bloom-causing...."

Page 13, line 17: Refer to Fig S3b rather than Fig 6b as the treatment-specific dynamics of nanophytoplankton observed during the early stage of the experiment (days 1-9) are visible in Fig S3b but cannot be clearly resolved in Fig 6b.

Page 34, Table 2: Why are table columns the p-value data labelled "Day:"?

———————————————

---

## Author Comment (AC1) · 15 May 2020

Reviewer 1:

General comments

This work is part of a minicosm investigation of the effects of increasing fCO2 levels on a natural planktonic microbial community of Prydz Bay, East Antarctica, and deals with the response of heterotrophic flagellates (HNF), nano- and picophytoplankton, and prokaryotes. The design of the experiments was similar to that of previous studies in East Antarctica, but with an initial CO2 acclimation period. The publications (one of

them, at least, in Biogeosciences) on the same minicosm experiment, and will have benefitted from the reviews of the previous works.

Overall, the manipulations appear to have been competently carried out and the text is well written. Concerning the discussion, I appreciated, in particular, the consideration given to potential community shifts, in addition to physiological changes. Some comments on aspects that could be improved are given below.

Specific comments

The main results of these accompanying works tend to appear late in the text; they should rather be presented up front in the introduction, so that the reader can better appreciate what is the context for and the contribution of the present study.

*Response: We agree that presenting the previously published results of this minicosm study in the Introduction will provide greater context for the results presented. We will update the Introduction to include a summary of the previously published findings of this minicosm study.*

Some conclusions go further than supported by the presented results. For example, the statement (whether correct or not): "Therefore, it is likely that increasing CO2 will cause the phytoplankton community to shift from a summer community that is currently dominated by large diatoms to one composed of smaller species or morphotypes of nano- and picophytoplankton." (lines 27-29 of page 13) does not derive from the work shown in the present manuscript (or if the authors believe so, it should be much better discussed).

*Response: This is true, we did not analyse the phytoplankton community >50 $\mu m$*

*in size so cannot solely base this conclusion on the results of this manuscript. This conclusion took into account the additional data provided by microscopic analysis of the microphytoplankton community in Hancock et. al (2018). We will reconsider this conclusion in its current location and will update this section of text to be more specific to the work in the present manuscript. We will provide further discussion of the combined published results of the greater minicosm study in the Conclusion.*

Other comments

It would be helpful for the readers to give more details on the statistical analyses (for example, explain "l" in tables S2-S5, number of time points and of pseudoreplicates).

*Response: We regret not having provided sufficient information regarding the statistical analysis. A number of changes will be made to the presentation and interpretation of statistical analyses (see also other referee comments below) and more clarification will be provided regarding pseudoreplicate numbers and abbreviations displayed in statistical tables.*

It would be helpful to repeat somewhere that the prokaryote group here is supposed to include few or no cyanobacteria.

*Response: The referee is entirely correct, the prokaryote analysis is of the heterotrophic prokaryote community only. This is because autotrophic prokaryotes (ie, cyanobacteria) were not detected in our study. We did mention in the Introduction that cyanobacteria are very rare in coastal Antarctic waters but we will reiterate this information in the Methods to make it abundantly clear that they were not detected in our flow cytometry results and were not part of the prokaryote analysis.*

Line 3 of page 9. Eliminate "treatments".

*Response: We will fix this sentence to remove the extra word.*

Lines 6-7 of page 10. "acclimating cells over the years to decades . . . is unachievable in most experimental designs". It is also doubtful to expect that the same cells/taxa would be acclimating for years or decades in natural settings.

*Response: This is true and we will amend the wording of this sentence to acknowledge this.*

Lines 7-8 of page 13. "dominated by large diatoms and ..." Which were the main "large diatom taxa"?

*Response: Previous observational studies of East Antarctic waters, of which the study site is located, identified a diverse range of large diatom taxa (e.g. Davidson et. al, 2010). The most abundant during summer were generally Fragilariopsis sp., Chaetoceros sp., Thalassiosira sp., Navicula sp., and Pseudo-nitzschia sp. In our minicosm study, the dominant species were large centric and pennate diatoms such as Thalassiosira sp. and Fragilariopsis sp. (see Hancock et. al, 2018). We will update this sentence to specify that we are discussing East Antarctic phytoplankton communities and provide examples of the dominant large diatom genera in early summer in this region.*

Lines 28-29 of page 13. "a summer community that is currently dominated by large

diatoms". This would not apply to many Antarctic areas.

*Response: We disagree with this statement. Summer blooms of large diatoms have been observed frequently across East Antarctic coastal regions and the Antarctic Peninsula (e.g. Ducklow et. al, 2007, Davidson et. al, 2010). The Ross Sea is one region where this is not necessarily the case and where large blooms of Phaeocystis antarctica are observed during the summer months (Arrigo et. al, 2000). That said, we do acknowledge that the driving factors for community composition differ around Antarctica. Our experiment was performed in East Antarctic waters and we did not intend our statement to encompass all Antarctic waters. We will update our conclusions to make this clearer.*

Line 31 of page 13. "Increases of prokaryote .."

*Response: We will fix this.*

Explanation of Fig. 7: "prokaryotes" instead of "prokryotes".

*Response: We will fix this.*

Explanation of Fig. 9: Add indication that the abscisssa shows the picophytoplankton and prokaryote abundances on the day before decline. For example: "Heterotrophic nanoflagellate abundance (y axis) on the day before (a) picophytoplankton and (b) prokaryote abundance (shown in x axis) declined in each ... "

*Response: We will revise the figure explanation to clarify the identity of the axes.*

References:

Arrigo, K.R., DiTullio, G.R., Dunbar, R.B., Robinson, D.H., VanWoert, M., Worthen, D.L., Lizotte, M.P., 2000. Phytoplankton taxonomic variability in nutrient utilization and primary production in the Ross Sea. J. Geophys. Res. Ocean. 105, 8827–8846. https://doi.org/10.1029/1998JC000289

Davidson, A.T., Scott, F.J., Nash, G. V., Wright, S.W., Raymond, B., 2010. Physical and biological control of protistan community composition, distribution and abundance in the seasonal ice zone of the Southern Ocean between 30 and 80°E. Deep Sea Res. Part II Top. Stud. Oceanogr. 57, 828–848. https://doi.org/10.1016/j.dsr2.2009.02.011

Ducklow, H.W., Baker, K., Martinson, D.G., Quetin, L.B., Ross, R.M., Smith, R.C., Stammerjohn, S.E., Vernet, M., Fraser, W., 2007. Marine pelagic ecosystems: the West Antarctic Peninsula. Philos. Trans. R. Soc. B Biol. Sci. 362, 67–94. https://doi.org/10.1098/rstb.2006.1955

Hancock, A.M., Davidson, A.T., McKinlay, J., McMinn, A., Schulz, K.G., van den Enden, R.L., 2018. Ocean acidification changes the structure of an Antarctic coastal protistan community. Biogeosciences 15, 2393–2410. https://doi.org/10.5194/bg-15-2393-2018

---

## Author Comment (AC2) · 15 May 2020

Reviewer 2:

General Comments

Heterotrophic nanoflagellates play and important role in pelagic marine ecosystems as grazers of picoplankton and as prey for microheterotrophs (ciliates and heterotrophic dinoflagellates). Polar marine ecosytems are especially vulnerable to ocean acidification and a several studies have investigated the effects of Ocean acidification on Antarctic and Arctic pelagic microbial communities. This paper reports

the results of a study which employed experimental minicosms to examine the effects of ocean acidification on pelagic microbial communities in Antarctica coastal waters. It contributes new observations on predator-prey interactions in response to acidification and supports observations derived from a previous study undertaken the same location using a similar experimental approach. The novelty of the study, as compared to the previous study, lies in the incorporation of an initial acclimation period within the experimental design. It is also useful and somewhat novel to encounter a paper which repeats and reinforce the insights gained from earlier work. The paper is well presented with a methods and data interpretation are results. However, there are some weaknesses in data interpretation and conclusion, outlined below, which should be addressed. In summary, the paper should make a valuable contribution to the literature addressing a timely and relevant topic which should be of interest to readers of Biogeochemistry.

Specific Comments

Section 2.4 (Page 5, line 19): State volume of the single sample removed from each minicosm on each day for flow cytometry analysis (from which different sets of pseudoreplicates were subsequently removed for analysis of different microbial groups).

*Response: We will update the flow cytometry methods to include the volume information.*

Section 2.4 (Page 5, line 22): Flow cytometry is the main method used to generate microbial data in the study so the authors should describe in full how flow cytometer sample flow rates were calibrated. This is important as flow cytometer flow rates are highly variable, and the exact volume analysed from each sample must be assessed

independently. Poor calibration technique is therefore a significant source of error in some studies. Describing how flow rates were calibrated gives confidence that the microbial abundance data are accurate. It also reminds readers that rigorous procedures are required to generate accurate data from flow cytometers.

*Response: We agree that understanding the flow cytometry methods is important. The "high" and "low" flow rates for each flow cytometer was calibrated by performing a linear regression of sample volume analysed by increasing time increments (in mins). This allowed us to determine the volume analysed for each different assay, based on the run time and flow speed setting. We appreciate the reviewer drawing our attention to the calibration data, as in doing so we found an error in the calculation of the pico- and nanophytoplankton abundance after changing to the FACSCalibur instrument on Day 16. This error was applied across all treatments so did not affect the overall CO2 treatment trends, but it did reduce the abundance observed on days 16-18. We have taken this error very seriously and have re-analysed all of our data to ensure that our results and conclusions have not changed. In the updated manuscript we will include more detail in the Methods to explain how the flow rate and volume calibrations were performed and put the data into a table in the Supplement. We will also update the individual flow cytometry methods for each group to specify the exact flow rates and volumes for each assay so that the calculations are clearer. Lastly, we will include an explanation that we had to use a different flow cytometer on day 16 because the FACScan broke down. All figures and tables will be remade to ensure they display the corrected data.*

Section 2.4.3 (Page 6, line 18): The prokaryote abundance measurements (undertaken by flow cytometry using SYBR Green I stain and FL1 versus SSC plots) will include phototrophic prokaryotes (i.e. picophytoplankton) as well as heterotrophic prokaryotes, unless the picophytoplankton data (derived from analyses in Section 2.4.1) were

subtracted from the prokaryote counts. This should be stated in this section of the methods. Heterotrophic prokaryotes will dominate these data, especially in Antarctic waters, so it is acceptable to treat the prokaryote results as representing mainly heterotrophic bacteria in subsequent discussions.

*Response: Autotrophic prokaryotes (ie, cyanobacteria) are rare in the region where we performed our experiment (see Wright et. al, 2009) and we did not observe any in our flow cytometry analysis. We have mentioned this in the Introduction but we will include this information in the Methods as well to make it clearer that the prokaryote data included only heterotrophic species.*

Section 2.5 (Page 7, Line 3): The authors correctly state that "statistical differences among treatments should be interpreted conservatively" due to lack of true replication. Clear trends between treatments can be clearly identified over the duration of incubation. However, conclusions based on the analysis of statistically significant differences between treatments (based on pseudoreplicates) at any one time point (page 7, line 1) are unconvincing. These include the subsequent statements (page 8, line 7; page 9, line 5) based on the picophytoplankton and prokaryote peak abundance analyses shown in Figure 7. Also the comparison of picophytoplankton and prokaryote growth rates with heterotrophic nanoflagellate abundance (page 9, line 20) shown in Figure 8. The respective conclusions from these analyses (that picoplankton and prokaryote abundance differ between treatments, and that reduced heterotrophic nanoflagellate abundance reduced grazing on picoplankton) are well-supported by the other analyses using data from several time points. I therefore question whether the authors should include the analyses presented in Figures 7 and 8.

Sections 3.4 (Page 8, line 15) and Section 4.2 (page 13, lines 12 and 23): I see no evidence in Fig 6b or Fig S3b that nanophytoplankton abundance was higher than the control in the 954 uatm treatment. The modelling data (shown in Table 2) may have

revealed this but the modelling result is a simulation of the underlying data which, in turn, is based on pseudoreplicates. The figures clearly show that nanophytoplankton abundance was higher than the control in the 634 treatment, but the case for the 954 uatm treatment resulting in higher abundance is unconvincing.

*Response: Upon reviewing the results, we agree there is not a good case for higher abundance of the 953 μatm treatment in the nanophytoplankton community. This result was based on a model that we accept was not well-fitted to the abundance of the treatments. Based on the above feedback of the statistical analysis in Sections 2.5 & 3.4 and our correction of pico- and nanophytoplankton abundance on days 16-18, we are reassessing our statistical methods. Examples of this are providing more robust modelling of growth curves through the use of generalized additive models (GAMs) to assess temporal changes in the abundance of the various microbial groups and removal of single time point analyses.*

Section 4.1 (Page 11, line 10): I am not convinced of the utility of the conclusion that heterotrophic nanoflagellate communities may change by 2050 due to ocean acidification. The abundance and composition of Antarctic heterotrophic nanoflagellate communities may well change by 2050 for many reasons, and microcosm experiments undertaken over 18 days cannot simulate real environmental changes to entire ecosystems over decades. I suggest this conclusion is removed.

*Response: We understand and have acknowledged in the Discussion that simulating real environmental changes to ecosystems over decades is a limitation of our experimental design. We also acknowledge that changes in CO2 are one of a number of environmental factors that will influence these communities with climate change (see Deppeler and Davidson, 2017). We will reconsider the wording of this conclusion and supply caveats around the onset and magnitude of additional stressors that may affect the HNF community response to ocean acidification.*

Section 4.1 (Page 11, line 13): The discussion on top-down control of heterotrophic nanoflagellates by the microheterotrophic community (heterotrophic dinoflagellates and ciliates) could include some additional considerations, as follows:

First, the study by Hancock et al. (2018) assessed microheterotroph abundance in Lugol's fixed samples of 2 to 10ml volume. It is not possible to derive meaningful microheterotroph data from such small sample volumes, so the statement (page 11, line 14) that treatments had no effect on the heterotrophic dinoflagellates and ciliates may not be valid.

*Response: The reviewer has misinterpreted the methods for Lugol's-fixed micro-heterotroph analysis in Hancock et. al (2018). The 2-10 ml samples were sedimented concentrates of seawater that were derived from 960 ml of sea water. Hancock et. al (2018) did, however, acknowledge that microheterotroph abundance was low ( 1% of all cells) and that a lack of CO2 response may have been related to these low counts. We shall therefore add that the response may not have been apparent due to the low abundance of these species in the experiment.*

Second, it would be useful to discuss the evidence for any switching in grazing pressure by microheterotrophs between nanophytoplankton and heterotrophic nanoflagellates. The fact that nanophytoplankton abundance was similar between treatments (except for 634 uatm – Figure 6b) suggests that heterotrophic dinoflagellates and ciliates were not exhibiting differential grazing pressure on heterotrophic nanoflagellates between treatments. This, in turn, lends support to the conclusion that the lower heterotrophic nanoflagellate abundances in high CO2 treatments were not a result of top-down pressure (assuming microheterotroph numbers were not affected by acidification
and similar in each treatment). On the other hand, the observed shift in the composition of the nanophytoplankton community from Phaeocystis to Fragilariopsis in high $CO_2$ treatments (page 13, line 21), as reported by Hancock et al. (2018), suggests that one would expect some differential microheterotrophic grazing between treatments and possible switching between nanophytoplankton and heterotrophic nanoflagellate prey.

*Response: This is an interesting consideration and we will re-evaluate our results and include discussion on possible changes in grazing pressure by microheterotrophs and heterotrophic nanoflagellates. Low abundances of heterotrophic dinoflagellates and ciliates in all treatments does suggest that grazing pressure on HNF was low and that reductions in heterotrophic nanoflagellate abundance at higher $CO_2$ levels were not likely caused by increased grazing from larger taxa. We will also consider in the Discussion how a shift in the dominant nanophytoplankton taxa might affect grazing dynamics.*

Third, the consequences of screening the seawater used to fill the minicosm tanks through a 200 micron filter should be discussed. This action will have reduced topdown grazing pressure on microheterotrophs, possibly creating a differential trophic cascade effect between treatments over the 18 days incubation. Any such effects may well have been minimal and equal across treatments. However, the potential effect of initial seawater screening should be discussed, especially with respect to the limits to which minicosm experiments can simulate the dynamics of in situ communities.

*Response: This is also an interesting consideration and we will include discussion on how a reduction in top-down pressure of larger zooplankton species may have affected the results. We routinely pre-screen the microbial community by 200 $\mu$m in these experiments because small differences in the abundance of large grazers among tanks could greatly affect the trajectory and composition of the succession in the tanks, thereby masking any $CO_2$-induced effect. We do appreciate that grazing of*

*>100% of daily production is observed in waters in this region (see Pearce et al. 2010). For this study, pre-screening by <200 $\mu$m allowed us greater control by only varying one environmental factor so we could focus on the effect of CO2 on the microbial community dynamics. We will include some consideration around how this may have affected our results in the Discussion.*

Section 4.1 (Page 12, line 5): Mixotrophic nanoflagellates will have been included within the nanophytoplankton counts due to the presence of chlorophyll (albeit possibly at low levels) within the cells. This should be mentioned in the methods or discussion text.

*Response: This is true, we still lack a thorough understanding of the mixotrophic community so we cannot comment on what proportion of nanophytoplankton cells are mixotrophic species. We will provide acknowledgement that mixotrophs will be part of the nanophytoplankton data. In addition to this, we will clarify that chlorophyll-containing mixotrophic cells will only be present among the nanophytoplankton but are not included in the heterotrophic nanoflagellate counts due to our removal of chlorophyll-containing cells as a first step to identifying heterotrophic cells.*

Section 4.3 (Page 13, line 32) and Section 4.4 (page 14, line 34): The results of Westwood et al. (2018) should be discussed as they are derived from the same location and draw similar conclusions to the present study (i.e. enhanced bacterial production and abundance in high CO2 treatments coinciding with reduced heterotrophic nanoflagellate abundance).

*Response: A comprehensive analysis of the results of Westwood et. al (2018) in relation to bacterial production in this minicosm study has been provided in Deppeler*

*et. al (2018). However, we appreciate that the findings of Westwood et. al (2018) provide valuable support to the current paper and will include further analysis and comparison with the results reported by them in our Discussion.*

Technical Corrections

Page 1, Title: The title could be misinterpreted as reporting the effects of ocean acidification on the "grazing of heterotrophic nanoflagellates." by their microzooplankton predators. A more accurate but unwieldly wording would be "reduces growth of and grazing on heterotrophic nanoflagellates.". Perhaps rephrase as "reduces growth and grazing impact of heterotrophic nanoflagellates".

Page 2, line 6: Correct spelling "whish" to "which".

Page 2, line 27: Use of the phrase "in the present study" implies that the observations referred to are part of the submitted manuscript rather than a different publication. Perhaps use the phrase "concurrently observed amongst choanoflagellates in the present minicosm experiment"?

Page 6, line12: The text refers to Figure 2a which shows a plot of FL3 versus FSC, rather than FL3 versus FL2 as stated in text.

Page 9, Line 5: "Fig. 7" should read "Fig. 7b".

Page 11, line 27: Add hyphen to change text to "bloom-causing"

Page 13, line 17: Refer to Fig S3b rather than Fig 6b as the treatment-specific dynamics of nanophytoplankton observed during the early stage of the experiment (days 1-9) are visible in Fig S3b but cannot be clearly resolved in Fig 6b.

Page 34, Table 2: Why are table columns the p-value data labelled "Day:"?

*Response: We agree with all the above technical corrections kindly provided by the*

*reviewer and will amend the text accordingly. In response to the final comment about the column label "Day:", the "Day:506" specifies the interaction term for the variables "Day" and "506" (i.e., CO2 treatment) in the statistical model. So, this is the result of the response of the CO2 treatment over time. We will make a number of changes to the presentation and interpretation of the statistical analyses (see above comments), so the information provided in text and in the tables is clear to the reader.*

References:

Deppeler, S.L., Davidson, A.T., 2017. Southern Ocean Phytoplankton in a Changing Climate. Front. Mar. Sci. 4, 1–18. https://doi.org/10.3389/fmars.2017.00040

Deppeler, S., Petrou, K., Schulz, K.G., Westwood, K., Pearce, I., McKinlay, J., Davidson, A., 2018. Ocean acidification of a coastal Antarctic marine microbial community reveals a critical threshold for CO2 tolerance in phytoplankton productivity. Biogeosciences 15, 209–231. https://doi.org/10.5194/bg-15-209-2018

Hancock, A.M., Davidson, A.T., McKinlay, J., McMinn, A., Schulz, K.G., van den Enden, R.L., 2018. Ocean acidification changes the structure of an Antarctic coastal protistan community. Biogeosciences 15, 2393–2410. https://doi.org/10.5194/bg-15-2393-2018

Pearce, I., Davidson, A.T., Thomson, P.G., Wright, S., van den Enden, R., 2010. Marine microbial ecology off East Antarctica (30 - 80°E): Rates of bacterial and phytoplankton growth and grazing by heterotrophic protists. Deep Sea Res. Part II Top. Stud. Oceanogr. 57, 849–862. https://doi.org/10.1016/j.dsr2.2008.04.039

Westwood, K.J.J., Thomson, P.G.G., van den Enden, R.L.L., Maher, L.E.E., Wright, S.W.W., Davidson, A.T.T., 2018. Ocean acidification impacts primary and bacterial production in Antarctic coastal waters during austral summer. J. Exp. Mar. Bio. Ecol. 498, 46–60. https://doi.org/10.1016/j.jembe.2017.11.003

Wright, S.W., Ishikawa, A., Marchant, H.J., Davidson, A.T., van den Enden, R.L., Nash, G. V., 2009. Composition and significance of picophytoplankton in Antarctic waters. Polar Biol. 32, 797–808. https://doi.org/10.1007/s00300-009-0582-9
* * *